# Using Imperfect Surrogates for Downstream Inference: Design-based Supervised Learning for Social Science Applications of Large Language Models

**Naoki Egami**[*1], **Musashi Hinck**[2], **Brandon M. Stewart**[* 2], **Hanying Wei**[1]

[1]Columbia University, [2]Princeton University

## Abstract

In computational social science (CSS), researchers analyze documents to explain social and political phenomena. In most scenarios, CSS researchers first obtain labels for documents and then explain labels using interpretable regression analyses in the second step. One increasingly common way to annotate documents cheaply at scale is through large language models (LLMs). However, like other scalable ways of producing annotations, such surrogate labels are often imperfect and biased. We present a new algorithm for using imperfect annotation surrogates for downstream statistical analyses while guaranteeing statistical properties—like asymptotic unbiasedness and proper uncertainty quantification—which are *fundamental* to CSS research. We show that direct use of surrogate labels in downstream statistical analyses leads to substantial bias and invalid confidence intervals, even with high surrogate accuracy of 80–90%. To address this, we build on debiased machine learning to propose the *design-based supervised learning* (DSL) estimator. DSL employs a doubly-robust procedure to combine surrogate labels with a smaller number of high-quality, gold-standard labels. Our approach guarantees valid inference for downstream statistical analyses, even when surrogates are arbitrarily biased and without requiring stringent assumptions, by controlling the probability of sampling documents for gold-standard labeling. Both our theoretical analysis and experimental results show that DSL provides valid statistical inference while achieving root mean squared errors comparable to existing alternatives that focus only on prediction without inferential guarantees.

## 1 Introduction

Text as data—the application of natural language processing to study document collections in the social sciences and humanities—is increasingly popular. Supervised classifiers have long been used to amortize human effort by scaling a hand-annotated training set to a larger unannotated corpus. Now large language models (LLMs) are drastically lowering the amount of labeled data necessary to achieve reasonable performance which in turn sets the stage for an increase in supervised text as data work. Recent papers have shown that GPT models can (for some tasks) classify documents at levels comparable to non-expert human annotators with few or even no labeled examples [Brown et al., 2020, Gilardi et al., 2023, Ziems et al., 2023].

In social science, such text classification tasks are only the first step. Scholars often use labeled documents in downstream analyses for *explanation* of corpus level properties [Hopkins and King, 2010, Egami et al., 2022, Feder et al., 2022, Grimmer et al., 2022], for example, using a logistic regression to model a binary outcome $Y \in \{0, 1\}$ by regressing this outcome on some explanatory

---

[*]Correspondence to `naoki.egami@columbia.edu` and `bms4@princeton.edu`. We provide the supplementary materials in this link (`https://naokiegami.com/paper/dsl_supplement.pdf`).

37th Conference on Neural Information Processing Systems (NeurIPS 2023).

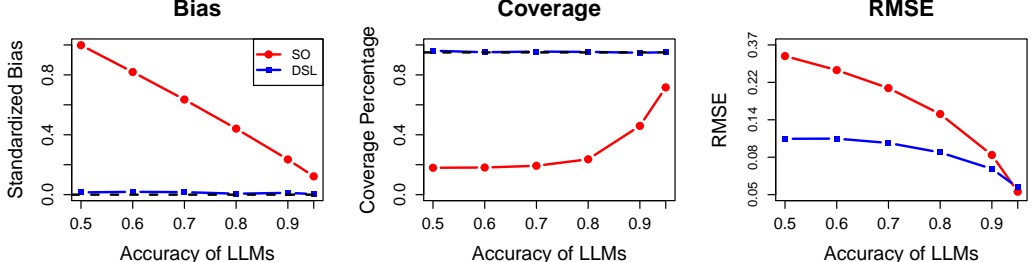

(a) **Simulated performance of Surrogate-Only Estimation (SO) and DSL**. Even for highly accurate surrogates, ignoring measurement error leads to non-trivial bias and undercoverage of 95% confidence intervals in downstream regression. Correct coverage and asymptotic unbiasedness are essential properties for proper uncertainty quantification—a must in social science. These concerns are resolved using DSL. The data-generating process uses a logistic regression similar to the one in Vansteelandt and Dukes [2022] and is described in the supplement.

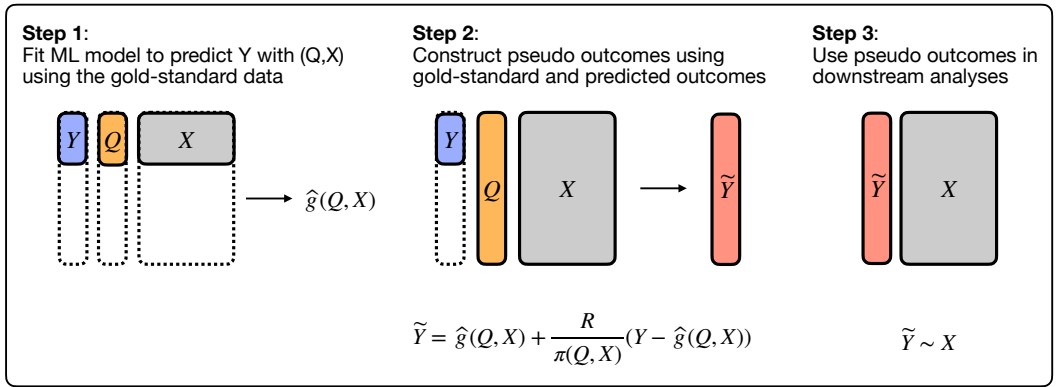

(b) **The DSL Estimator** $Y$ represent gold-standard outcomes available only for a subset of documents. $Q$ represent surrogate labels (e.g. from an LLM). $X$ represent explanatory variables that social scientists use in downstream statistical analyses. $\widehat{g}(Q, X)$ is a supervised machine learning model to predict $Y$ with $(Q, X)$. In the second step, we construct pseudo-outcomes by combining gold-standard outcomes $Y$, predicted outcomes $\widehat{g}(Q, X)$, an indicator variable for gold-standard labeling $R$ (taking 1 if hand-coded and 0 otherwise), and the known probability of gold-standard labeling $\pi(Q, X)$. In the final step, researchers use pseudo-outcomes in downstream statistical analyses, e.g., regressing $\widetilde{Y}$ on $X$. A full notation summary is available in Table 2.

Figure 1: **Overview of the Problem and Design-based Supervised Learning (DSL)**

variables $X \in \mathbb{R}^{d_X}$. In political science, $Y$ could represent whether a social media post contains hate speech, and $X$ could include posters' characteristics, such as gender, education, and partisanship. Regression estimates the share of hate speech posts within levels of explanatory variables. Importantly, this task of explanation in CSS is distinct from unit-level prediction—classifying whether each post contains hate speech. Because of this social science goal of explanation, simple regression models, like logistic regression, are often preferred as low dimensional summaries [Chernozhukov et al., 2018b, Vansteelandt and Dukes, 2022]. Ideally, all documents would be labeled by experts to achieve a gold-standard $Y$ but this is costly and so researchers turn to more scalable approaches such as LLMs. We call these more scalable—but error-prone—labels, *surrogate labels*.

When using surrogates in downstream statistical analyses, researchers often ignore measurement error—the unknown and heterogeneous mismatch between the gold-standard label and the surrogate [Knox et al., 2022]. Measurement error is ubiquitous in CSS research due to the inherent difficulty of the task [Ziems et al., 2023], unknown social and racial biases in LLMs [Zhao et al., 2018, Bender et al., 2021], and performance sensitivity to classifiers or prompt engineering [Perez et al., 2021, Zhao et al., 2021]. Ignoring measurement error leads to estimator bias and invalid confidence intervals in downstream statistical analyses even when the surrogate labels are extremely accurate. In a simulated example (shown in Figure 1-(a)), even with surrogate accuracy of 90%, the bias is substantial and the coverage of a 95% confidence interval is only 40% in downstream regression. This is a fatal flaw in the

| Method | Data Requirement | Statistical Properties | | |
|---|---|---|---|---|
| | No Gold-Standard | Consistency | Coverage | Low RMSE |
| Surrogate Only (SO) | ✓ | ✗ | ✗ | ? |
| Gold-Standard Only (GSO) | ✗ | ✓ | ✓ | ✗ |
| Supervised Learning (SL) | ✗ | ✗ | ✗ | ? |
| **Design-Based SL (DSL)** | ✗ | ✓ | ✓ | ? |

Table 1: **Design-based Supervised Learning (DSL) compared with existing approaches.** The statistical properties are compared under settings where researchers control the probability of sampling documens for gold-standard labeling, while no additional assumptions about fitted supervised machine learning models are made. RMSE is marked as indeterminate (?) since the relative order between SO, SL, and DSL depends on the amount of bias specific to applications.

social sciences, where estimated effects are generally small—such that even low measurement error can overturn scientific conclusions—and valid uncertainty quantification is *essential* to distinguish signal from noise.

We propose a method to use surrogate labels as outcomes for common statistical analyses in the social sciences while guaranteeing consistency, asymptotic normality and valid confidence intervals. We assume a setting where the analyst has a large collection of documents and is interested in a regression of some text-based outcome on some known document-level characteristics. We observe imperfect surrogate labels for all documents and can choose to sample a small number of documents with known probability for an expert to annotate yielding gold-standard labels. Our proposed estimator, *design-based supervised learning* (DSL), combines the surrogate and gold-standard labels to create *a bias-corrected pseudo-outcome*, which we use in downstream statistical analyses (see Figure 1-(b)). The proposed DSL estimator is consistent and asymptotically normal, and its corresponding confidence interval is valid, *without any further modeling assumptions even when the surrogates are arbitrarily biased*. While we do not require accurate surrogates, as their accuracy improves, the efficiency of the DSL increases. These strong theoretical guarantees only require that the sampling probability of the documents selected for gold-standard be known and be bounded away from zero. Both conditions are straightforward to guarantee by design in many social science applications where the whole corpus is available in advance, which gives the name, *design-based* supervised learning.

After describing our contributions and related work, we formally characterize the problem setting (Section 2). In Section 3, we describe existing approaches including (i) using the surrogates ignoring the measurement error, (ii) using only gold-standard labels and ignoring the surrogates, and (iii) the conventional supervised approaches which use both. In Section 4, we describe our proposed method and prove its theoretical properties. Section 5 benchmarks our method against existing approaches in 18 diverse datasets, demonstrating that DSL is competitive in root mean squared errors while consistently delivering low bias and proper coverage. See Table 1 for a summary. Section 6 concludes with a discussion of limitations.

**Contributions.** We propose a unified framework for using imperfect surrogate labels in downstream statistical analyses which maintains the CSS priority for unbiasedness and proper coverage. We exploit three features of text-labeling tasks in social science: researchers often control the probability of sampling documents for gold-standard labeling, the accuracy of surrogate labels will likely keep increasing, and most quantities of interest can be written as a regression (i.e. not requiring individual labels). We (1) provide a new estimator, (2) prove strong theoretical guarantees, including consistency and proper asymptotic coverage, without requiring the accuracy of the surrogate or the correct specification of the underlying supervised machine learning estimator, (3) offer extensions to a broad range of moment-based estimators, and (4) demonstrate finite sample performance across 18 datasets which leverage LLMs for surrogate annotation. Our framework provides a theoretically-sound, design-based strategy for downstream analyses.

**Related Work.** In the text as data literature, there is limited work on addressing measurement error in regressions with predicted outcomes [although see, Bella et al., 2014, Wang et al., 2020, Zhang, 2021]. Existing approaches use a separate gold-standard sample to estimate the error rate and perform a post-hoc correction to the regression. This is related to approaches for calibrating a classifier [Platt, 1999, Zhang, 2021]. The related literature on *quantification* seeks to characterize

the share of documents in each class and thus corresponds to the intercept-only regression model with a categorical outcome [Forman, 2005, González et al., 2017]. The quantification literature has historically combined this task with domain shift since otherwise the mean of the training data is an extremely strong baseline [Hopkins and King, 2010, Keith and O'Connor, 2018, Card and Smith, 2018, Jerzak et al., 2023]. Our approach encompasses the quantification problem without domain shift (an intercept-only regression) and can handle cases where quantification is used to model different subgroups of the data that are available at training time (using regression).

Our proposed method also draws upon the large literature on double/debiased machine learning and doubly-robust estimation for missing data and causal inference [Robins et al., 1994, Laan and Robins, 2003, Chernozhukov et al., 2018a, Kennedy, 2022]. In particular, our use of bias-corrected pseudo-outcomes builds on foundational results on semiparametric inference with missing data [Robins and Rotnitzky, 1995, Tsiatis, 2006, Rotnitzky and Vansteelandt, 2014, Davidian, 2022] and the growing literature on doubly robust estimators for surrogate outcomes [Kallus and Mao, 2020] and semi-supervised learning [Chakrabortty and Cai, 2018, Chakrabortty et al., 2022]. A similar framework of using doubly robust estimation to debias measurement errors has also been recently used in other application areas [Angelopoulos et al., 2023, Mozer and Miratrix, 2023]. Like these papers, we exploit the efficient influence function to produce estimators with reduced bias.

Finally, we join an increasing number of papers that explore a variety of different uses of large language models for social science questions [Ornstein et al., 2022, Gilardi et al., 2023, Velez and Liu, 2023, Wu et al., 2023, Ziems et al., 2023]. Our work, in particular, focuses on how to correct biases and measurement errors in outputs from large language models in order to perform valid downstream statistical analyses. We also contribute to the growing literature on using predicted variables in downstream statistical analyses in the social sciences [Fong and Tyler, 2021, Knox et al., 2022, Katsumata and Yamauchi, 2023].

## 2 The Problem Setting and Design-based Sampling

Consider the case where a researcher wants to classify documents into a binary outcome $Y \in \{0, 1\}$ and then regress this outcome on some explanatory variables $X \in \mathbb{R}^{d_X}$ using a logistic regression (we consider extensions to non-binary outcomes and more general moment-based estimators below).

Suppose we have $n$ independent and identically distributed samples of documents. For all documents, we observe surrogate labels $Q \in \mathbb{R}^{d_Q}$, optional additional meta-data about the documents, $W \in \mathbb{R}^{d_W}$, which might be predictive of $Y$, and explanatory variables $X$ to be included in our regression. We assume the outcome is costly to measure and we can choose to have an expert annotate a subset of the documents to provide the gold-standard $Y$. We use a missing indicator $R_i \in \{0, 1\}$ to denote whether document $i$ is labeled by experts ($R_i = 1$) or not ($R_i = 0$). Therefore, we observe the data $\{R_i, R_i Y_i, Q_i, W_i, X_i\}_{i=1}^n$, and the total number of gold-standard documents is given by $n_R = \sum_{i=1}^n R_i$. We use $\pi(Q_i, W_i, X_i) \coloneqq \Pr(R_i = 1 \mid Q_i, W_i, X_i)$ to denote the probability of sampling document $i$ for gold-standard labeling. Formally, we assume that the sampling probability for gold-standard labeling is *known* and bounded away from zero.

**Assumption 1 (Design-based Sampling for Gold-Standard Labeling)**
*For all $i$, $\pi(Q_i, W_i, X_i)$ is known to researchers, and $\pi(Q_i, W_i, X_i) > 0$.*

The assumption holds when the researcher directly controls the sampling design. For example, if a researcher has 10000 documents and samples 100 of them to expert-annotate at random, $\pi = \frac{100}{10000} = .01$ for all documents. We also allow more complex stratified sampling schemes (shown later in our applications) and can cover any case where the sampling depends on the surrogates, optional covariates or, explanatory variables such that $\pi(Q_i, W_i, X_i)$ is known. Restricting ourselves to this sampling mechanism allows us to guarantee, $Y \perp\!\!\!\perp R \mid Q, W, X$. This assumption does rule out some applications—such as instances of domain shift where documents from the target population are not available at annotation time—but captures most social science research applications where researchers need to annotate a corpus of documents which is available in total from the outset.

Our estimand of interest is the coefficients of the oracle logistic regression $\beta^* \in \mathbb{R}^{d_X}$, which is a solution to the following moment equations.

$$\mathbb{E}\{(Y - \text{expit}(X^\top \beta))X\} = 0, \tag{1}$$

| | |
|---|---|
| $Y_i$ | Outcome, which is observed for documents labeled by experts. |
| $X_i$ | Explanatory variables we include in downstream regression analysis. |
| $\beta$ | Coefficients of the downstream regression analysis. Our main estimand of interest. |
| $Q_i$ | Surrogate outcome (e.g., LLM annotation). |
| $W_i$ | Optional covariates that are predictive of $Y$. |
| $R_i$ | Missing indicator, indicates whether we sample document $i$ for gold-standard labeling. |
| $\pi(Q_i, W_i, X_i)$ | Prob. of sampling document $i$ for gold-standard labeling depending on $(Q_i, W_i, X_i)$. |
| $\widehat{g}(Q_i, W_i, X_i)$ | Estimated supervised machine learning model predicting $Y$ as a function of $(Q_i, W_i, X_i)$. |

Table 2: **Summary of Our Notation.**

where expit($\cdot$) is the inverse of the logit function. $\beta^*$ is defined as the solution to the moment equations above. Here, $\beta^*$ is seen as a low-dimensional summary, and thus, this paper does not assume the underlying data-generating process follows the logistic regression.

## 3 Existing Approaches: Their Methodological Challenges

Because we do not observe $Y$ for all documents, we cannot directly solve equation (1) to estimate $\beta^*$. This motivates the three existing strategies that researchers use in practice: using only the surrogate, using only the subset of documents that have gold-standard annotations, and using a conventional supervised learning approach. None of these approaches attain asymptotically unbiased estimation with proper coverage under minimal assumptions while also using surrogate labels to increase efficiency.

**Surrogate Only Estimation (SO)**    The most common approach in practice is to replace $Y$ with one of the surrogate labels ignoring any error. While we have motivated this with LLM-generated labels, this can be generated by any previously trained classifier, an average of labels produced from multiple LLMs/prompts, or any other strategy not using the gold-standard data. For example, researchers can construct LLM-based text label $Q_i$ as a surrogate for the outcome $Y_i$ in each document $i$, and then they can run a logistic regression regressing $Q_i$ on $X_i$.

The appeal of this approach is that it uses all documents $n$ for downstream analyses. This method is consistent with valid confidence intervals only when measurement errors are random and mean-zero conditional on $X$, which is rarely the case in practice. Recent papers have evaluated LLMs and concluded that the predictions are 'accurate enough' to use without correction in at least some settings [Ornstein et al., 2022, Ziems et al., 2023, Törnberg, 2023, Gilardi et al., 2023]. However as we saw in Figure 1-(a) even substantially more accurate LLM predictions can lead to non-trivial error.

**Gold-Standard Only Estimation (GSO)**    Even if the researcher wants to use a surrogate, they presumably produce gold-standard annotations for a subset of documents to evaluate accuracy. The simplest approach of obtaining valid statistical inference is to use only this gold-standard data, ignoring documents that only have a surrogate label. Regressing $Y_i$ on $X_i$ only using gold-standard data with weights $1/\pi(Q_i, W_i, X_i)$ is equivalent to solving the following moment equations,

$$\sum_{i:R_i=1} \frac{1}{\pi(Q_i, W_i, X_i)}(Y_i - \text{expit}(X_i^\top \beta))X_i = 0, \tag{2}$$

where the summation is taken only over documents with $R_i = 1$. Regular M-estimation theory shows that the estimated coefficients are consistent and their corresponding confidence intervals are valid [van der Vaart, 2000]. The key limitation of this approach is that it ignores the surrogate labels which, while biased, can help improve efficiency.

**Supervised Learning (SL)**    Some researchers combine gold-standard data and LLMs-based surrogate outcomes in supervised learning [e.g., Wang et al., 2020, Zhang, 2021]. While the exact implementation varies, we consider a most common version of this strategy where researchers fit a black-box supervised machine learning model to predict $Y$ with $(Q, W, X)$ using the gold-standard data. Then, using the fitted supervised machine learning model, $\widehat{g}(Q_i, W_i, X_i)$, researchers predict labels for the entire documents and use the predicted label as the outcome in downstream logistic

regression. This is equivalent to solving the moment equation,

$$\sum_{i=1}^{n}(\widehat{g}(Q_i, W_i, X_i) - \text{expit}(X_i^\top \beta))X_i = 0. \tag{3}$$

Researchers can combine this with sample-splitting or cross-fitting to avoid overfitting bias. This estimation method is consistent only when $g(Q_i, W_i, X_i)$ is correctly specified and consistent to the true conditional expectation $\mathbb{E}(Y \mid Q, W, X)$ in $L_2$ norm, i.e., $||\widehat{g}(Q, W, X) - \mathbb{E}(Y \mid Q, W, X)||_2 = o_p(1)$ as sample size $n$ goes to infinity. This assumption is trivial when the surrogate is binary and there are no covariates, but implausible in more general settings. More problematically, in general, this estimator cannot provide valid confidence intervals due to regularization bias, even when the underlying machine learning model is correctly specified [Chernozhukov et al., 2018a].

What we call SL here is a broad class of methods. It includes as special cases the surrogate only estimator ($\widehat{g}(Q_i, W_i, X_i) = Q_i$) and classical supervised learning with crossfitting (here there is no $Q_i$ and we predict using other document features $W_i$ and $X_i$). With appropriate modification, it also includes post-hoc corrections using a gold-standard set as in Levy and Kass [1970], Hausman et al. [1998], Wang et al. [2020], Zhang [2021]. All of these strategies are expected to perform well in terms of RMSE. However, for these methods to be consistent, they need to assume the correct specification of the estimator for $g$, which is often implausible in social science applications. Even under such an assumption, they do not provide valid confidence intervals or p-values, unless other additional stringent assumptions are imposed.

## 4 Our Proposed Estimator: Design-based Supervised Learning

No existing strategy meets our requirements of asymptotically unbiased estimation with proper coverage while also efficiently using the surrogate labels. Design-based supervised learning (DSL) improves upon the conventional supervised learning procedure (which is not generally consistent and does not provide valid confidence intervals) by using a bias-corrected pseudo-outcome in downstream statistical analyses (summarized in Definition 1 and Algorithm 1).

For the proposed DSL estimator, we employ a $K$-fold cross-fitting procedure [Chernozhukov et al., 2018a]. We first partition the observation indices $i = 1, \ldots, n$ into $K$ groups $\mathcal{D}_k$ where $k = 1, \ldots, K$. We then learn the supervised machine learning model $\widehat{g}_k$ by predicting $Y$ using $(Q, W, X)$ using all expert-coded documents *not* in $\mathcal{D}_k$. We then define a bias-corrected pseudo-outcome $\widetilde{Y}_i^k$ for observations in $\mathcal{D}_k$ as follows.

$$\widetilde{Y}_i^k := \widehat{g}_k(Q_i, W_i, X_i) + \frac{R_i}{\pi(Q_i, W_i, X_i)}(Y_i - \widehat{g}_k(Q_i, W_i, X_i)). \tag{4}$$

This pseudo-outcome can be seen as the sum of the predicted labels $\widehat{g}_k(Q_i, W_i, X_i)$ (the same as in the conventional supervised learning) and the bias-correction term $\frac{R_i}{\pi(Q_i, W_i, X_i)}(Y_i - \widehat{g}_k(Q_i, W_i, X_i))$. Our use of the pseudo-outcome builds on a long history of the doubly robust methods [e.g., Robins et al., 1994, Rotnitzky and Vansteelandt, 2014, Chakrabortty et al., 2022].

This bias-correction step guarantees that the conditional expectation $\mathbb{E}_k(\widetilde{Y}^k \mid Q, W, X)$ is equal to the true conditional expectation $\mathbb{E}_k(Y \mid Q, W, X)$ under Assumption 1, even when the supervised machine learning estimator $\widehat{g}$ is misspecified. Here we use $\mathbb{E}_k$ to denote the expectation over $\mathcal{D}_k$, which is independent of data used to learn $\widehat{g}_k$ in cross-fitting.

$$\begin{aligned}
\mathbb{E}_k(\widetilde{Y}^k \mid Q, W, X) &:= \mathbb{E}_k\left(\widehat{g}_k(Q, W, X) + \frac{R}{\pi(Q, W, X)}(Y - \widehat{g}_k(Q, W, X)) \middle| Q, W, X\right) \\
&= \frac{\mathbb{E}_k(RY \mid Q, W, X)}{\pi(Q, W, X)} + \left(1 - \frac{\mathbb{E}_k(R \mid Q, W, X)}{\pi(Q, W, X)}\right)\widehat{g}_k(Q, W, X) \\
&= \mathbb{E}_k(Y \mid Q, W, X)
\end{aligned}$$

where the first line follows from the definition of the pseudo-outcome and the second from the rearrangement of terms. The final line follows because $\mathbb{E}_k(RY \mid Q, W, X) = \mathbb{E}_k(R \mid Q, W, X)\mathbb{E}_k(Y \mid Q, W, X)$ based on Assumption 1, and $\mathbb{E}_k(R \mid Q, W, X) = \pi(Q, W, X)$ by definition. Importantly, this equality does not require any assumption about the supervised machine learning method $\widehat{g}_k(Q, W, X)$ or about measurement errors. The proposed DSL estimator exploits this robustness of the bias-corrected pseudo-outcome to the misspecification of $\widehat{g}(Q, W, X)$.

**Definition 1 (Design-based Supervised Learning Estimator)** *We first construct the bias-corrected pseudo-outcome $\widetilde{Y}_i^k$ as in equation (4) using K-fold cross-fitting. Then, we define the design-based supervised (DSL) estimator for logistic regression coefficient $\beta$ to be a solution to the following moment equations.*

$$\sum_{k=1}^{K} \sum_{i \in \mathcal{D}_k} (\widetilde{Y}_i^k - \text{expit}(X_i^\top \beta)) X_i = 0. \tag{5}$$

---

**Algorithm 1** Design-based Supervised Learning

---

**Inputs:** Data $\{R_i, R_i Y_i, Q_i, W_i, X_i\}_{i=1}^n$, Known gold-standard probability $\pi(Q_i, W_i, X_i)$ for all $i$.
**Step 1:** Randomly partition the observation indices into $K$ groups $\mathcal{D}_k$ where $k = 1, \ldots, K$.
**Step 2:** Learn $\widehat{g}_k$ from gold-standard documents not in $\mathcal{D}_k$ by predicting $Y$ with $(Q, W, X)$
**Step 3:** For documents $i \in \mathcal{D}_k$, construct the bias-corrected pseudo-outcome $\widetilde{Y}_i^k$ (see equation (4))
**Step 4:** Solving the logistic regression moment equation by replacing $Y_i$ with $\widetilde{Y}_i^k$ (see equation (5))
**Outputs:** Estimated coefficients $\widehat{\beta}$ and Estimated variance-covariance matrix $\widehat{V}$

---

Proposition 1 (proof in supplement) shows that the DSL estimator is consistent and provides valid confidence intervals when the gold-standard probability is known to researchers, without requiring the correct specification of the supervised machine learning method.

**Proposition 1** *Under Assumption 1, when the DSL estimator is fitted with the cross-fitting approach (Algorithm 1), estimated coefficients $\widehat{\beta}$ are consistent and asymptotically normal.*

$$\sqrt{n}(\widehat{\beta} - \beta^*) \xrightarrow{d} \mathcal{N}(0, V) \tag{6}$$

*In addition, the following variance estimator $\widehat{V}$ is consistent to $V$, that is, $\widehat{V} \xrightarrow{p} V$, where*

$$\widehat{V} \ := \ \widehat{\mathbf{M}}^{-1} \widehat{\Omega} \widehat{\mathbf{M}}^{-1}, \quad \widehat{\mathbf{M}} := \frac{1}{n} \sum_{i=1}^n \text{expit}(X_i^\top \widehat{\beta})(1 - \text{expit}(X_i^\top \widehat{\beta}))) X_i X_i^\top,$$

$$\widehat{\Omega} \ := \ \frac{1}{n} \sum_{k=1}^K \sum_{i \in \mathcal{D}_k} (\widetilde{Y}_i^k - \text{expit}(X_i^\top \widehat{\beta}))^2 X_i X_i^\top.$$

This proposition highlights the desirable theoretical properties of DSL relative to alternatives. First, the estimators of coefficients are consistent, and asymptotic confidence intervals are valid. These results hold even when $g$ is arbitrarily misspecified as long as it does not diverge to infinity. This is unlike the SL approach which requires that $g$ is correctly specified and is consistent to the true conditional expectation $\mathbb{E}(Y|Q, W, X)$. Second, even when LLMs-based labels are arbitrarily biased, such biased measures do not asymptotically bias final estimates of coefficients $\beta$. Therefore, unlike the SO approach, researchers can use LLMs-based labels and retain theoretical guarantees even when there is a concern of differential measurement errors. Finally, the asymptotic variance $V$ decreases with the accuracy of the surrogate labels, allowing for use of non-gold-standard data unlike GSO.

These powerful theoretical guarantees mainly come from the *research design* where researchers know and control the expert-coding probability $\pi(Q, W, X)$. The well known double/debiased machine learning framework mostly focuses on settings where the expert-coding probability $\pi$ is unknown and needs to be estimated from data. In such settings, researchers typically need to assume (i) estimation of nuisance functions, such as $g$ and $\pi$, is correctly specified and consistent to the true conditional expectations, and (ii) they satisfy particular convergence rates, e.g., $o_p(n^{-0.25})$. We do not require either assumption because we exploit the research design where the expert-coding probability $\pi(Q, W, X)$ is known.

**Extension: Method of Moment Estimator** Our proposed DSL estimator can accommodate any number of surrogate labels and a wide range of outcome models that can be written as a class of moment estimators with what we will call *design-based moments*. Importantly, many common estimators in the social sciences, including measurement, linear regression, and logistic regression, can be written as the method of moment estimator. In general, suppose researchers are interested in a method of moment estimator with a moment function $m(Y, Q, W, X; \beta, g)$ where $(Y, Q, W, X)$

are the data, $\beta$ are parameters of interest, and $g$ is the supervised machine learning function. Then, the estimand of interest $\beta_M^*$ can be written as the solution to the following moment equations. $\mathbb{E}(m(Y, Q, W, X; \beta, g^*)) = 0$, where $g^*$ is the true conditional expectation $\mathbb{E}(Y \mid Q, W, X)$. We define the moment function to be *design-based* when the moment function is insensitive to the first step machine learning function.

**Definition 2 (Design-based Moments)** *A moment is design-based when* $\mathbb{E}(m(Y, Q, W, X; \beta, g)) = \mathbb{E}(m(Y, Q, W, X; \beta, g'))$ *for any* $\beta$ *and any machine learning functions* $g$ *and* $g'$ *that do not diverge.*

Note that the design-based moment is most often a doubly robust moment, and Chernozhukov et al. [2022] provide a comprehensive theory about doubly robust moments. In this general setup, the DSL estimator $\widehat{\beta}_M$ is a solution to the following moment equation.

$$\sum_{k=1}^{K} \sum_{i \in \mathcal{D}_k} m(Y_i, Q_i, W_i, X_i; \beta, \widehat{g}_k) = 0. \tag{7}$$

**Proposition 2** *Under Assumption 1, when the DSL estimator with a design-based moment is fitted with the cross-fitting approach,* $\widehat{\beta}_M$ *is consistent and asymptotically normal.*

We provide the proof and the corresponding variance estimator in the appendix.

## 5 Experiments

In this section, we verify that our theoretical expectations hold in 18 real-world datasets. We compare DSL and all three existing approaches, demonstrating that only DSL and GSO meet the standard for bias and coverage, while DSL improves efficiency. We use generalized random forests via `grf` package in `R` to estimate $g$ function [Tibshirani et al., 2022] and all cross-fitting procedures use five splits. A comparison to Wang et al. [2020] is included in the supplement.

**Logistic Regression in Congressional Bills Data** We use data from the Congressional Bills Project [CBP, Adler and Wilkerson, 2006] to construct a benchmark regression task. CBP is a database of 400K public and private bills introduced in the U.S. House and Senate since 1947. Each bill is hand-coded by trained human coders with one of 20 legislative topics. Our downstream analysis is a logistic regression to examine the association between whether a bill is labeled `Macroeconomy` ($Y$) and three traits of the legislator proposing the bill ($X$)—whether the person is a senator, whether they are a Democrat, and the DW-Nominate measure of ideology [Lewis et al., 2023]. For our simulations, we use 10K documents randomly drawn from documents labeled with the `Macroeconomy` (the positive class), or the `Law and Crime`, `Defense` and `International Affairs` topics (reflecting that the negative class is often diverse). We consider two scenarios: in the balanced condition, there are 5K documents in each class. In the imbalanced condition, there are 1K documents in the positive class and 9K documents in the negative class. For surrogate labels ($Q$), we include zero-shot predictions using GPT-3 [Brown et al., 2020] which achieves accuracy of 68% (balanced) and 90% (imbalanced). As an additional document-covariate ($W$), we include the cosine distance between embeddings of the document and the class description using MPnet [Reimers and Gurevych, 2019, Song et al., 2020]. Additional details and experiments using five-shot predictions are included in the appendix.

Figure 2 compares the performance of our four estimators on bias, coverage and RMSE. As expected, only GSO and DSL perform well on bias and achieve nominal coverage. While SL is able to achieve better RMSE at these sample sizes, DSL achieves a 14% gain in RMSE over GSO (balanced condition) as shown in Figure 3. This increases to 31% in the five-shot results. The empirical result accords with the theory: only DSL and SO provide a low-bias estimator achieving nominal coverage and DSL is notably more efficient.

**Class Prevalence Estimation** Ziems et al. [2023] evaluate zero-shot performance of a variety of LLMs on 24 diverse CSS benchmark tasks including detection tasks for emotion, hate-speech, ideology and misinformation. They, like others [e.g. Ornstein et al., 2022, Gilardi et al., 2023] make a qualified recommendation for zero-shot classification only (our SO estimator) in research-grade settings noting that "In some lower-stakes or aggregate population analyses, 70% [accuracy] may be a sufficient threshold for direct use in downstream analyses" [Ziems et al., 2023, p. 13]. We evaluate

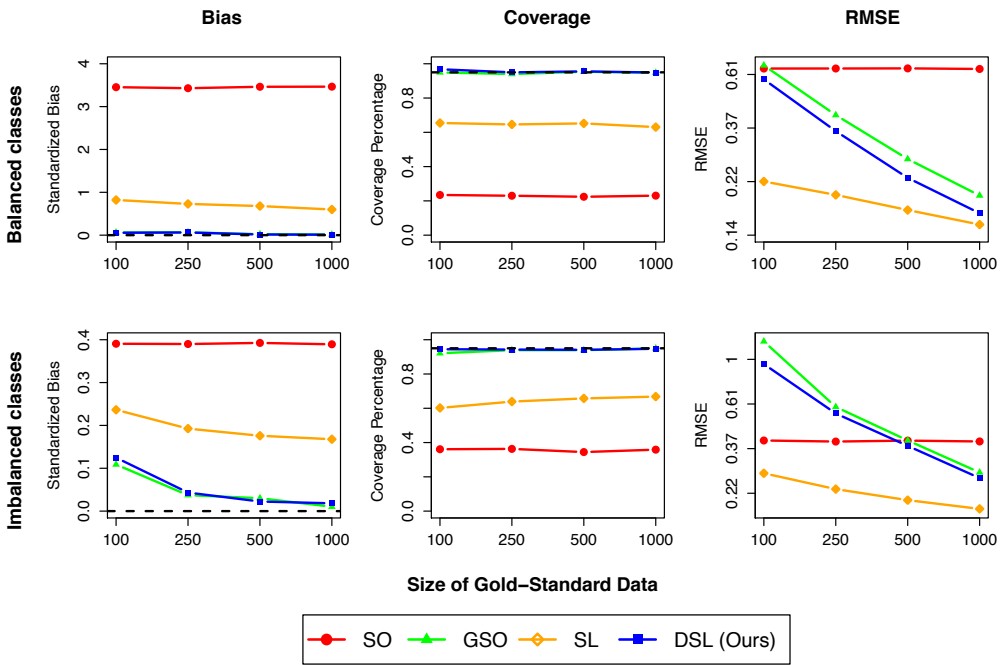

Figure 2: **Logistic regression estimation with Congressional Bills Project Data.** Results for a three variable logistic regression model of a binary outcome indicating whether a bill is about `Macroeconomy`. Bias shows the standardized root mean squared bias (averaged over the three coefficients). Coverage shows proportion of 95% confidence intervals covering the truth. RMSE plots the average RMSE of the coefficients on a log scale. Each sampled dataset contains 10K datapoints with the X-axis providing gold-standard sample size. We average over 500 simulations at each point. Only DSL and GSO are able to achieve proper coverage, but DSL is more efficient.

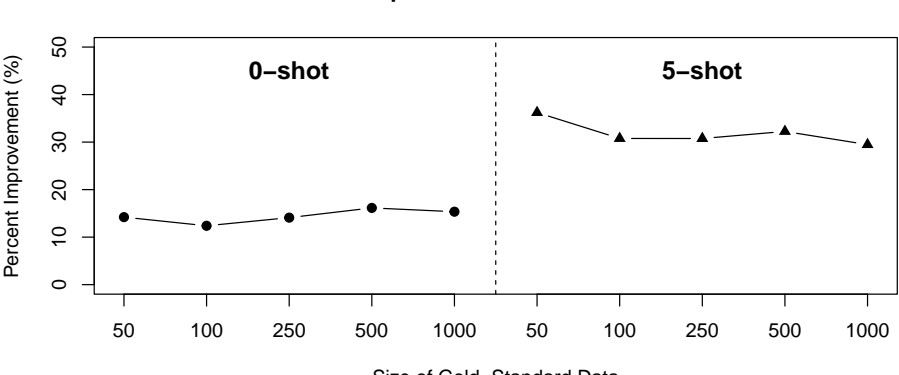

Figure 3: **Improvement of DSL over GSO.** Both DSL and GSO attain asymptotic unbiasedness and proper coverage. Here we show the gain in efficiency for DSL over GSO in the balanced condition. As the quality of the surrogate rises (here as we move from the 0-shot to 5-shot setting) the efficiency gain from DSL grows.

on the 17 datasets in their Table 3 using `flan-ul2` [Chung et al., 2022] as the surrogate which is one of the highest performing LLMs in their study. Because there are no consistent covariates for these studies, we focus on class prevalence estimation—the simplest case of our regression setting.

| Dataset | | LLM | | Bias (×100) | | | | Coverage (Percentage) | | | | RMSE (×100) | | | |
| (# cat.) | obs. | Acc. | F1 | SO | GSO | SL | DSL | SO | GSO | SL | DSL | SO | GSO | SL | DSL |
|---|---|---|---|---|---|---|---|---|---|---|---|---|---|---|---|
| Misinfo. (2) | 500 | 77.6 | 77.4 | 9.0 | 0.0 | 0.2 | 0.0 | 0.6 | 94.8 | 95.2 | 96.2 | 9.2 | 5.0 | 4.3 | 4.3 |
| Emotion (6) | 498 | 70.3 | 70.1 | 2.0 | 0.1 | 0.6 | 0.1 | 67.2 | 94.2 | 91.3 | 94.0 | 2.9 | 3.7 | 3.2 | 3.2 |
| Figur. (4) | 500 | 64.0 | 61.7 | 5.5 | 0.2 | 0.3 | 0.2 | 41.2 | 94.1 | 91.8 | 94.3 | 6.0 | 4.4 | 3.8 | 3.9 |
| Power (2) | 500 | 60.8 | 57.8 | 26.6 | 0.0 | 0.2 | 0.1 | 0.0 | 94.8 | 93.2 | 94.8 | 26.6 | 5.0 | 5.0 | 4.9 |
| Toxic. (2) | 500 | 56.6 | 50.5 | 34.8 | 0.0 | 0.4 | 0.0 | 0.0 | 94.8 | 93.8 | 94.8 | 34.9 | 5.0 | 5.1 | 4.9 |
| Disc. (7) | 497 | 41.9 | 39.5 | 5.4 | 0.1 | 0.5 | 0.1 | 39.7 | 93.7 | 91.4 | 93.8 | 5.8 | 3.5 | 3.4 | 3.4 |
| News (3) | 498 | 40.3 | 38.8 | 10.6 | 0.0 | 0.3 | 0.0 | 4.3 | 95.1 | 94.0 | 95.1 | 10.8 | 4.7 | 4.6 | 4.5 |
| Emp. (3) | 498 | 39.8 | 34.7 | 24.7 | 0.0 | 0.2 | 0.0 | 0.0 | 95.1 | 93.7 | 94.9 | 24.8 | 4.7 | 4.8 | 4.7 |
| Hate (6) | 498 | 35.9 | 32.8 | 9.8 | 0.1 | 0.4 | 0.1 | 17.4 | 94.2 | 92.6 | 94.1 | 10.1 | 3.7 | 3.6 | 3.6 |

Table 3: **Class prevalence estimation on a subset of the 17 datasets from Ziems et al. [2023] with $n = 100$ gold-standard labels.** Multi-class tasks are converted to 1-vs-all binary tasks with reported performance averaged over tasks. SO, SL and DSL use a surrogate label from `flan-ul2`. Numbers in green indicate any estimator within 0.1pp of the lowest bias; blue indicate any estimator achieving above 94.5% coverage; orange indicate any estimator achieving within 0.001 of the best RMSE. Remaining 8 tasks are shown in the supplement.

Table 3 shows the results for $n = 100$ gold-standard labels (analyses at different $n$ in the supplement) ordered by the accuracy of the LLM surrogate. As with the logistic regression our theoretical expectations hold. Even for surrogate labels above 70% accuracy, the SO estimator has high bias and consequentially poor coverage for this aggregate quantity. Using only 100 gold-standard annotations, DSL is able to achieve very low bias and nominal (or near nominal) coverage with consistent gains in RMSE compared to GSO (the only other estimator to attain acceptable bias and nominal coverage). Importantly, the RMSE of DSL is also comparable to SL.

## 6   Discussion

We have introduced a design-based supervised learning estimator which provides a principled framework for using surrogate labels in downstream social science tasks. We showed competitive performance across 18 diverse CSS classification tasks with LLM surrogates, providing low bias and approximately nominal coverage, while achieving RMSE comparable to existing alternatives that focus only on prediction without inferential guarantees. Our approach works for any predicted outcome used in a downstream task [see e.g. Wang et al., 2020, for several examples across fields].

**Limitations.**   We briefly describe three limitations of our work. First, DSL is focused on a very specific setting: outcome variables used in a regression where researchers need to annotate a corpus of documents *available from the outset*. This is a common setting in social science, but not a universal one: we might use text as a predictor [Fong and Tyler, 2021, Katsumata and Yamauchi, 2023], require individual document classifications, or have domain shift in the target population. Second, you need a way to construct gold-standard labels. This will often be naturally accessible using data that might otherwise be used to evaluate accuracy of the classifier. Like any method using gold-standard labels, DSL can be sensitive to the gold-standard being accurate—something which is not always true even in competition test sets and might not even be feasible in some settings. Finally, our method focuses on social science research settings where the researcher's priority is bias and coverage rather than RMSE. As SL explicitly targets RMSE, it may be the better option if that is the primary consideration. We often find that the RMSE tradeoff is small in order to have low bias and nominal coverage.

**Future Work.**   In future work we plan to explore the entire analysis pipeline including improvements to prompt engineering and the implications of conditioning prompt engineering on preliminary assessments of performance. This paper assumed the probability of gold-standard labeling is given, but a natural future extension is to consider optimal sampling regimes for the gold-standard data.

**Acknowledgments**   We are very grateful to Amir Feder for excellent feedback on an earlier draft. Research reported in this publication was supported by the Department of Political Science at Columbia University, the Data-Driven Social Science Initiative at Princeton University, and Princeton Research Computing.

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
