# OpenReview forum: "Using Imperfect Surrogates for Downstream Inference: Design-based Supervised Learning for Social Science Applications of Large Language Models"
_NeurIPS.cc/2023/Conference — NeurIPS 2023 poster_

### Official Review · Reviewer_rbu9 · 2023-06-25

**Soundness:** 3 good
**Presentation:** 3 good
**Contribution:** 2 fair
**Rating:** 6
**Confidence:** 4

**Summary:**

The author presents a complete and thorough study in their research regarding semi-supervised learning of social science application targeting bias and coverage control. The work include clear mathematical proof of the purposed approach, and experimental results demonstrating the well alignment of theory and application. The assumption is also clearly stated and limitations are considered. As a non-expert of computational social science, the paper is relatively easy to read and understand. Overall I think this is a solid work in the field. I have a few minor questions and comments below.

**Strengths:**

1. the mathematical proof of bias correction step, equation 4 and the conditionally unbiased estimation of Yi is simple and clear
2. the surrogate labels can be arbitrarily biased, while the theoretical guarantee still holds is the most fascinating and important contribution of the work. I personally find this really interesting
3. experimental results show great alignment of theory, comparable performance of RMSE compared with SSL although the method does not specifically target for it

**Weaknesses:**

1. the biggest miss/weakness after reading from my perspective is from the title, using LLM annotations for valid statistical inference. In general the research is more like a mathematical proof of bound on semi-supervised learning and demonstration of usefulness with experiment. There is little regarding using LLM annotations. With such a title I would expect a substantial part of the work focusing on LLM generated annotations, e.g. how it is used and how can it be improved from LLM perspective, etc.
2. the assumption of the purposed approach, though clearly stated, greatly limit the contribution of the approach. It may be a common setting in social science that test set is known, but not the case for majority of LLM annotation use cases.
3. the asymptotic behavior given the gold-standard annotation, is not well discussed in the main paper. Aspects like the downstream coefficients expectation and variance given the size of golden labels compare with sample space size are not well covered. Some empirical numbers could be extremely useful, beyond the settings in experiment part
4. some minor comments: page 8 figure 2 annotations, as a non expert of social science it would be good to explain in more detail what is coverage and why it is important to the study. In second experiment class prevalence estimation, better include more details like each dataset's distribution, total size of each dataset, etc. (see it in supplement however I think it is helpful to include some numbers in main paper, as it gives the important size ratio of golden-set and total set)

**Questions:**

Please see comments in weakness part

**Limitations:**

the limitations of the work is well discussed.

---

> ### Author Rebuttal · Authors · 2023-08-08
>
> Thank you for the review! We are glad you found the theoretical and empirical work in the paper enjoyable!
>
> **Clarify the RMSE gain of DSL over GSO:**
>
> First, based on suggestions from reviewers, we further clarified the RMSE gains of our estimator compared to existing alternatives. In particular, we show that the RMSE gains of DSL compared to GSO (the only other method that is unbiased and has valid confidence intervals) are substantial: it is equivalent to having 50% more hand-coded documents (when using zero-shot LLM learning) and to having 100% more hand-coded documents (when using five-shot LLM learning). Please see the new Figure R-1 in the attached PDF. Given the high cost of hand-coding, this practically means that computational social scientists can obtain unbiased estimates and valid confidence intervals with much fewer hand-coded documents when they use DSL instead of existing alternatives. We wanted to emphasize the substantial efficiency gains from our method, which our previous Figure 2 did not convey well.
>
> **We quote the weaknesses below and respond.**
>
> **(1)** *the biggest miss/weakness after reading from my perspective is from the title, using LLM annotations for valid statistical inference. In general the research is more like a mathematical proof of bound on semi-supervised learning and demonstration of usefulness with experiment. There is little regarding using LLM annotations. With such a title I would expect a substantial part of the work focusing on LLM generated annotations, e.g. how it is used and how can it be improved from LLM perspective, etc.*
>
> Following your and other reviewers’ advice, we are changing the title to “**Using Imperfect Surrogates for Downstream Inference: Design-based Semi-supervised Learning for Social Science Applications of Large Language Models.**” We will also emphasize the generality of DSL and downplay the specific LLM angle in the paper. Please see more detailed responses in our Global Response (at the top of this page).
>
> **(2)** *the assumption of the purposed approach, though clearly stated, greatly limit the contribution of the approach. It may be a common setting in social science that test set is known, but not the case for majority of LLM annotation use cases.*
>
> We completely agree and this is why we have tried to be so clear both about the social science connection and the assumption. We are adding more citations of applications to the paper to make clear that this covers a sizable portion of social science use cases. We also want to further clarify that our method only requires a random subset of the test data (i.e., we do not need the full test data when we hand-code documents). An interesting future direction is to incorporate the domain shift (i.e., researchers do not have access to a random subset of the test data). We hope the reviewer agrees that the use case is broad enough to support the work, but we agree on the limitation.
>
> **(3)** *the asymptotic behavior given the gold-standard annotation, is not well discussed in the main paper. Aspects like the downstream coefficients expectation and variance given the size of golden labels compare with sample space size are not well covered. Some empirical numbers could be extremely useful, beyond the settings in experiment part*
>
> Thank you for raising this point! We want to clarify that this was indeed our goal in Figure 2 (which shows asymptotics for the CBP application). Bias is the difference between the expected values of estimated coefficients and the true coefficients, while RMSE is a summary measure of bias and variance. To make these points clearer, we created a new figure (Figure R-4 in the attached PDF) that directly visualizes the expected values of coefficients and variance. Please let us know whether the newly added figure is what you are thinking about. We are also happy to provide any additional detail you would like to see. In our new simulation results, we also consider a wide range of sample sizes (see Figure R-3 in the attached PDF).
>
> **(4)** *some minor comments: page 8 figure 2 annotations, as a non expert of social science it would be good to explain in more detail what is coverage and why it is important to the study.*
>
> Thank you for your question. The coverage is a probability that the 95% confidence interval covers the true coefficients over the sampling distribution. If an estimator has valid uncertainty quantification and valid confidence interval, this coverage should be (asymptotically) at least 95%. This valid uncertainty quantification is essential because social scientists often conduct statistical analyses to *test social science hypotheses* rather than making predictions. For example, when studying online hate speech, social scientists would be interested in testing whether highly educated people are less likely to post hate speech. To conduct such hypothesis testing, valid uncertainty quantification (with confidence intervals and corresponding p-values) is fundamental. We will provide this discussion in the paper and extend the caption of Figure 2.
>
> **(5)** *In second experiment class prevalence estimation, better include more details like each dataset's distribution, total size of each dataset, etc. (see it in supplement however I think it is helpful to include some numbers in main paper, as it gives the important size ratio of golden-set and total set)*
>
> Thank you for the suggestion! We amended Table 2 to include this information. Please see the updated Table 2 in the attached PDF. Please let us know whether the newly added table is what you are thinking about.

---

> > ### Author Response · Authors · 2023-08-18
> >
> > Please kindly let us know whether our responses addressed your questions and concerns. We are more than happy to add any additional analyses you think are necessary for the paper. We just wanted to ask this such that we can respond again to you within the discussion periods.

---

### Official Review · Reviewer_m9JA · 2023-06-25

**Soundness:** 3 good
**Presentation:** 3 good
**Contribution:** 3 good
**Rating:** 6
**Confidence:** 3

**Summary:**

In order to exploit the advantages of Large Language Models (LLMs) in computational social science (CSS) area, this paper proposed a novel DSL estimator, which employed a doubly-robust procedure to combine surrogate labels with a few gold-standard labels. By using the proposed method, not only LLMs can be used as surrogate label generator, but also the asymptotic unbiasedness and proper uncertainty quantification in CSS can be maintained. Finally, this paper conducted plenty of experiments to prove the effectiveness of the proposed method.

**Strengths:**

1.	This paper proposed a unified framework for using imperfect surrogate labels in downstream statistical analyses which maintains the CSS priority for unbiasedness and proper coverage.
2.	This paper proposed a strong theoretical guarantee for the effectiveness of the proposed method.
3.	This paper conducted extensive experiments across 18 datasets to compare the performance of the proposed method.


**Weaknesses:**

1.	This paper focused on the semi-supervised setting in some CSS scenarios. The difference between the proposed method and previous semi-supervised methods should be compared in a detailed way.
2.	This paper had plenty of notations. An explanation table about these notations should be provided.
3.	In experiments parts, this paper should provide a brief about the evaluation tasks and corresponding datasets, so that the results can be more convincing.


**Questions:**

N/A

**Limitations:**

Please see the weaknesses for more details.

---

> ### Author Rebuttal · Authors · 2023-08-08
>
> Thank you for the review! We are glad you enjoyed the theory and the extensive experiments.
>
> **Clarify the RMSE gain of DSL over GSO:**
>
> First, based on suggestions from reviewers, we further clarified the RMSE gains of our estimator compared to existing alternatives. In particular, we show that the RMSE gains of DSL compared to GSO (the only other method that is unbiased and has valid confidence intervals) are substantial: it is equivalent to having 50% more hand-coded documents (when using zero-shot LLM learning) and to having 100% more hand-coded documents (when using five-shot LLM learning). Please see the new Figure R-1 in the attached PDF. Given the high cost of hand-coding, this practically means that computational social scientists can obtain unbiased estimates and valid confidence intervals with much fewer hand-coded documents when they use DSL instead of existing alternatives. We wanted to emphasize the substantial efficiency gains from our method, which our previous Figure 2 did not convey well.
>
> **Below, we quote the weaknesses you identified below and respond.**
>
> **(1)** *This paper focused on the semi-supervised setting in some CSS scenarios. The difference between the proposed method and previous semi-supervised methods should be compared in a detailed way.*
>
> Thank you for this point. We have added an additional comparison with the best state of the art we are aware of (Wang et al. 2020 in *PNAS*), which, as far as we know, is the only published article that proposes an estimator that covers exactly the same settings as ours (downstream regression analyses). As this existing estimator requires stronger parametric assumptions, it is, in general, biased and has invalid confidence intervals in our experiments, in contrast to our proposed DSL. Please see our new results in Figure R-3 in the attached PDF. Importantly, we additionally show that, as sample size increases, DSL provably dominates SSL in terms of RMSE because bias of SSL does not vanish with sample size, while variance of DSL will vanish. See the same Figure R-3. We will also use the additional page to describe the results of our experiments more clearly and what they tell us about how DSL compares to prior SSL methods — essentially that they have slightly better RMSE, while all of the SSL methods are biased and have invalid confidence intervals. We are also open to any other details you would like to see. Thank you for your suggestions.
>
> **(2)** *This paper had plenty of notations. An explanation table about these notations should be provided.*
>
> Thank you for the suggestion. We will take the notation summary in Appendix 2.2 and append a condensed version to Figure 1, so readers will have an easily accessible notation guide.
>
> **(3)** *In experiments parts, this paper should provide a brief about the evaluation tasks and corresponding datasets, so that the results can be more convincing.*
>
> Thank you! Per the suggestion from you and one other reviewer, we will substantially extend our discussion of the details of the evaluations in the main paper by pulling in material from the appendix. In particular, we will be more clear about the specification of downstream analyses, the details of LLM annotations, and training procedures. We have also extended Table 2 to include more information about each dataset (please see a snippet of Table 2 in the attached PDF; we show two rows as an example due to the space constraint). If there are any other details you would find particularly essential to include, please let us know, and we are happy to accommodate. We think of the theoretical results as the most convincing piece of evidence and the experiments as demonstrating that the results hold in practice. The review has been helpful in clarifying that we need to bring in more details.

---

> > ### Author Response · Authors · 2023-08-18
> >
> > Please kindly let us know whether our responses addressed your questions and concerns. We are more than happy to add any additional analyses you think are necessary for the paper. We just wanted to ask this such that we can respond again to you within the discussion periods.

---

> > > ### Comment · Reviewer_m9JA · 2023-08-21
> > >
> > > The authors have addressed all my questions. I will keep my opinion unchanged.

---

> > > > ### Author Response · Authors · 2023-08-21
> > > >
> > > > We are glad that our responses addressed your questions. Thank you so much again for taking the time to read our paper and provide a review. We will make sure to incorporate these changes mentioned above, which will improve our paper.

---

### Official Review · Reviewer_qrXw · 2023-07-08

**Soundness:** 2 fair
**Presentation:** 2 fair
**Contribution:** 2 fair
**Rating:** 4
**Confidence:** 4

**Summary:**

This paper studies the use of surrogate labels in model training and it's bias, coverage and performance properties. Practically, the premise is to use a small number of gold data annotations for a classification task, then construct surrogate labels that have desirable bias, coverage and performance properties. The application of this would be doing additional analyses using covariates that would benefit from the additional support in data size.

The method proposed uses a small amount of gold labels to calibrate the model prediction and is agnostic to the choice of underlying model. The method is compared to using gold labels alone, surrogate labels alone from an external model not fitted to the data and a simple form of semi-supervised learning where a model is fitted to the gold labels.

Experiments are conduced in depth on bias, coverage and RMSE in more depth on one data set on congressional bills. Further experiments are conducted over a battery of data sets related to computational social science applications on bias, coverage and RMSE with 100 gold labels. The results show that bias and coverage for the proposed method are closer to using gold standard data alone, while showing small improvements on some datasets in RMSE.

**Strengths:**

The premise and application is interesting and has not been too well studied to the best of my knowledge.

A large number of data sets were used in the bias/coverage/RMSE experiments, which provides a good starting point for such analyses.

The proofs and experiments supporting that bias and coverage properties of the proposed method.

**Weaknesses:**

The key weakness is that application presented as a motivation for this paper is not tested throughout any of the experiments. The covariates are not used in any of the experiments to enhance data analysis, which could have been done with several data sets in the collection, including the congressional bills.

The experiments show minor improvements in general on the RMSE, especially when compared to the vanilla self-supervised methods. The self-supervised methods represented here are very weak for this category of methods. For example, the gold labels are overwritten by the self-supervision and the self supervision is done in a single round over the entire data set, without any filtering by confidence as usually done, which makes this an artificially weak baseline, especially on bias and coverage. Yet, the SS methods achieve constantly higher RMSE, especially when considering the difference between Gold labels and the proposed method (DSL), which shows that the DSL methods does not capture much of the performance benefits that self-supervision alone could achieve. This leaves the main application of the approach the ability to use more data for covariate analysis, but this application is now shown through experimentation in the paper.

The paper should also present results in F1 metric (both surrogate label model and method comparison results) because RMSE is not natural for multi-class classification and having the surrogate label model and methods trained on top of predictions on the same method is useful for comparisons.

As a general comment: the method is agnostic to the source of surrogate labels, as any model can be used to produce the labels and the proposed method does not use any information about the model or its assumptions. Hence, while LLMs can be a source for labels, the paper should be rewritten to remove so many specific mentions of LLMs and discussions about LLMs, which serve no purpose for the paper (perhaps it would for paper publicity), but actually artificially narrow the generality of the method.

The paper also cites many papers that make unfounded claims or estimates about accuracy needed for 'annotations' (surrogate labels) for applications or model training, while none of the cited papers (e.g. Gilardi 2023) actually trained models to test the improvements those could bring (or not).

**Questions:**

See weaknesses.

An interesting analysis would be to use either synthetic data or to intentionally vary the accuracy of the underlying model on the same data set (rather than existing accuracy which differs across data sets). This would provide a more controlled analysis of these factors.

What is the K (defined in line 192) in the experiments?

---

> ### Author Rebuttal · Authors · 2023-08-08
>
> Thank you for the review! We are glad that you enjoyed the proofs, experiments, and premise. Below, we quote the weaknesses you identified and respond.
>
> **(1)** *The key weakness is that the application presented as a motivation for this paper is not tested throughout any of the experiments. The covariates are not used in any of the experiments to enhance data analysis. [...]*
>
> We believe this might be a miscommunication due to our use of semisupervised learning (an overloaded term!). We first clarify our goal and then propose ways to incorporate your suggestions.
>
> Our goal is to *explain* a document label outcome using covariates with regression models (e.g., logistic regression). For example, political scientists are often interested in *explaining* what types of people are more likely to post hate speech. Here, outcome **Y** is whether a post contains hate speech, and covariates **X** could include posters’ characteristics (e.g. education and partisanship). This regression estimates the share of hate speech within strata of **X**. Social scientists run such statistical analyses to *test* social science hypotheses (e.g., whether highly educated people are less likely to post hate speech). These regressions are run explicitly or implicitly through the calculation of subgroup averages. This task of *explanation* and *hypothesis testing* in the social sciences is distinct from *unit-level prediction*—classifying whether each post contains hate speech. Document-level prediction is not less important; our goal is different. We developed DSL to perform downstream regression analyses that are asymptotically unbiased and have valid confidence intervals. Importantly the role of covariates is to use them as explanatory variables in the second-step downstream regression, not to improve model training for document-level predictions.
>
> We think your understanding was: use an LLM to annotate a set of documents, use those as the labels to train a supervised classifier, and then return this fine-tuned classifier as the output. This is similar to the common computer science goal of training a better model to predict labels at each document level. This goal is different from ours. Your concern, in this case, appears to be that we could have a stronger baseline by only training the classifier using labels about which the LLM is confident and potentially repeating over multiple rounds, folding in new examples each time. Please correct us in the discussion if we have misunderstood your point.
>
> These procedures are related but they tackle different goals. If the classifier trained on your procedure was used as the predicted labels in our model, it might yield a higher accuracy surrogate, but our main point is how to use such surrogates in downstream analyses. If we directly use such surrogates in the downstream regression, they will lead to substantial bias and invalid confidence intervals, even when the accuracy can be extremely high.
>
> You were concerned that we didn’t use the covariates “to enhance data analysis,” but the CBP example is specifically about the downstream regression of the Macroeconomy category regressed on three covariates in our downstream logistic regression. Thus, we believe that we have demonstrated in our experiments the core use case, and it is consistent with the vast majority of social science use cases. We will make this point clear in our revision and include additional citations.
>
> **(2)** *The experiments show minor improvements in general on the RMSE, especially when compared to the vanilla self-supervised methods. The self-supervised methods represented here are very weak for this category of methods. [...] Yet, the SS methods achieve constantly higher RMSE.*
>
> We want to clarify two points. First, we follow the social science priority for bias and coverage. SSL is biased and has invalid confidence intervals, in contrast to DSL. Second, focusing on RMSE, we expect that SSL, which is optimized for RMSE, can outperform DSL. However, in our experiment, we showed that DSL, while maintaining unbiasedness and valid confidence intervals, can achieve RMSE comparable to SSL (even though it is often higher than SSL in a finite sample). Importantly, we additionally show that, as sample size increases, DSL provably dominates SSL in terms of RMSE because bias of SSL does not vanish with sample size, while variance of DSL will vanish. See our new simulation evidence (Figure R-3 in the attached PDF). We have also added the state-of-the-art SSL baselines (Wang et al., *PNAS*), and found the same pattern (see the same Figure R-3).
>
> **(3)** *The paper should also present results in F1 metric [...].*
>
> We added F1 metric for the surrogate in Table 2 (see Table 2 in PDF). For our main results, the target is the coefficients of the downstream regression and the F1 metric does not apply.
>
> **(4)** *As a general comment: the method is agnostic to the source of surrogate labels [...].*
>
> Following your and other reviewers’ advice, we are changing the title to reflect this point. Please see the global response at the top of this page.
>
> **(5)** *The paper also cites many papers [...], while none of the cited papers [...] actually trained models to test the improvements those could bring (or not).*
>
> We think this is related to the miscommunication described above. Gilardi and others want to annotate documents so they can use LLM annotations as outcomes in downstream analyses without doing hand coding. We show this surrogate-only estimator is biased.
>
> **(6)** *An interesting analysis would be to use either synthetic data or to intentionally vary the accuracy of the underlying model on the same data set[...].*
>
> This was the goal of Figure 1a, and we now extended it by including additional simulations based on synthetic data (see Figure R-2 in PDF).
>
> **(7)** *What is the K in the experiments?*
>
> The number of splits which we set to five in both experiments (the default in Chernozhukov et al. 2018).

---

> > ### Author Response · Authors · 2023-08-18
> >
> > Please kindly let us know whether our responses addressed your questions and concerns. We are more than happy to add any additional analyses you think are necessary for the paper. We just wanted to ask this such that we can respond again to you within the discussion periods.

---

> > ### Comment · Reviewer_qrXw · 2023-08-21
> >
> > Hello,
> >
> > (1) I believe I understand and am in agreement with the statement in the response. The additional experiments presented in the response and not in the original paper, including the variation in performance of surrogate labels, do show evidence of this for this setup (although unclear if balanced or imbalanced). However, I believe a single test case (data set, data combination, selection of topics, data size, covariates, inclusion of the MPnet similarity not in the original paper) is not enough to prove general applicability, especially as the experimental setup is not standard for this data set.
> >
> > Separately, it seems there is a strange effect in the new charts that is not commented which could warrant some investigation: the RMSE is similar with varying the sample size from  50-1000 and the percent improvement is actually decreasing with larger sample size for 5-shot.
> >
> > Simply having more experiments across multiple data sets would lead to more robust insights.
> >
> > (2) The point is that the SS method is a weak baseline and it would add more weight to the paper to prove the bias and coverage is an issue for better SS methods.
> >
> > (3) The F1 should also be presented in the method comparison results, given many of the data sets are multi-class classification and/or using imbalanced data sets.
> >
> > (4) The change in wording I believe is unsatisfactory and would like to continue arguing that the insistence on the importance of LLM is not representative and restricting the applicability of the method.
> >
> > I find this comment a bit misplaced: "you already build up a gold standard dataset with which you can use DSL to improve performance" the gold standard is needed anyway to start from and the other parts of the response mention that DSL's main goal was actually not to improve performance.
> >
> > (5) The reference is primary about the 70% accuracy threshold copied from (Ziems at al 2023).
> >
> > (6) Thank you for adding the analysis in Figure R-2.

---

> > > ### Author Response · Authors · 2023-08-21
> > > **Response to Point 1**
> > >
> > > Thank you for your comments. Given the time constraint (we only had 12 hours between your response and the end of the discussion period), we were not able to produce additional experimental results. We addressed your numbered list below (thank you for continuing the numbering to keep our complex discussion well-organized!) with a short summary of the main points on each. Overall we are still concerned that there is a misunderstanding about the goal of our study.
> > >
> > > **Response to (1)**:
> > > To summarize the discussion thus far: Your original point (1) was to argue that the motivation for our paper was not tested in the experiments.  In our reply we argued that this was based on a miscommunication due to the variety of meanings given across fields for ‘semisupervised’ learning and restated our paper’s goals clarifying how the experiments target our use case.  In the response to our reply, you began with *I believe I understand and am in agreement with the statement in the response* and then proceeded to raise a few additional questions which we address below.
> > >
> > > Now we quote the rest of the reply in full and respond.
> > >
> > > *I believe I understand and am in agreement with the statement in the response. The additional experiments presented in the response and not in the original paper, including the variation in performance of surrogate labels, do show evidence of this for this setup (although unclear if balanced or imbalanced).*
> > >
> > > (1.1) In Figure 1 and Figure 2, we are showing the figures based on the balanced class but we have already done the same analyses for the imbalanced class and they show similar patterns.
> > >
> > > *However, I believe a single test case (data set, data combination, selection of topics, data size, covariates, inclusion of the MPnet similarity not in the original paper) is not enough to prove general applicability, especially as the experimental setup is not standard for this data set.*
> > >
> > > (1.2): Here we would respectfully point out that our original submission uses 18 different datasets.  For the rebuttal we had a space constraint of 1 page of figures and so we focused on showing the results for the CBP application (Figure R-1) and for a new simulation specifically designed to address your concerns (Figure R-2).  We are happy to commit to including similar figures for all 18 datasets in the appendix of the final version.  That said, we emphasize again that our primary evidence is the proof of the properties of DSL while we see the datasets as simply demonstrating that the proofs hold in real settings.  Finally there is a passing reference here to MPnet similarity.  We think this is a misinterpretation of the citation Wang et al.  The full citation for our reference is: Wang, Siruo, Tyler H. McCormick, and Jeffrey T. Leek. "Methods for correcting inference based on outcomes predicted by machine learning." *Proceedings of the National Academy of Sciences* 117.48 (2020): 30266-30275. This was cited in our paper and we included it as a baseline to address the concern you raised about absence of a strong baseline.  It is the strongest competing method targeting our task of which we are aware.
> > >
> > > *Separately, it seems there is a strange effect in the new charts that is not commented which could warrant some investigation: the RMSE is similar with varying the sample size from 50-1000 and the percent improvement is actually decreasing with larger sample size for 5-shot.*
> > >
> > > (1.3) Please note that the Y-axis on Figure R-1 is the percent improvement, so the absolute value of RMSE is in fact decreasing as the sample size increases. We can emphasize this point clearly in a caption when we include it in the appendix. Theoretically, the percent improvement converges to a certain number, and for the zero-shot case, we already see that the percent improvement converges to certain values (on the left panel, about 15% and, on the right panel, about 40-50%). As for the 5-shot case as well, we theoretically expect that this improvement will converge to a particular value. We can further increase the sample size and verify this in the final revised version.
> > >
> > > *Simply having more experiments across multiple data sets would lead to more robust insights*
> > >
> > > (1.4) As noted above, we are happy to provide similar analyses for all 18 different data sets in the final version.That said for a paper with a proof of the relevant property, we are comfortable with 18 datasets as sufficiently demonstrating that the properties of DSL are robust to a variety of real-world circumstances (including the range of tasks that Ziems et al use to represent computational social science).

---

> > > ### Author Response · Authors · 2023-08-21
> > > **Response to Points (2-3)**
> > >
> > > **Response to Point (2)**
> > > To summarize the discussion thus far: the original objection here was around the lack of RMSE gains over the semi-supervised methods (written as ‘self-supervised’ in the original review and part of our sense that there was a misunderstanding).  We replied by describing the importance of bias and coverage for social science and emphasizing that semi-supervised methods directly minimize RMSE.  We showed that we are able to match the RMSE of the Semi-Supervised Learning setup and further that we outperform the state-of-the-art baseline in the Wang et al paper cited above.
> > >
> > > Now we quote the rest of the reply in full and respond.
> > > *The point is that the SS method is a weak baseline and it would add more weight to the paper to prove the bias and coverage is an issue for better SS methods.*
> > >
> > > We are concerned that you might still be thinking about " **Self-Supervised** Learning" but we want to emphasize that our paper or our response never discuss " **Self-Supervised** Learning". As in the title and throughout the paper, we only talk about " **Semi-Supervised** Learning". While both are often abbreviated to be SSL, they are different methods, and our relevant alternative is not "Self-Supervised Learning". Our SSL method in the paper is consistently about " **Semi-Supervised** Learning," which is a class of methods that use the gold-standard labels to train the model and predict labels in the unlabeled data set. The central question is how you use predicted labels from any **Self-Supervised** learning methods in downstream analyses. If used directly in the downstream analyses, the estimator is "Surrogate-Only" and if it is used with additional fine-tuning with the gold-standard label, the estimator is "Semi-Supervised Learning".
> > >
> > > Your original review did not offer any citations and so we struggled to figure out what stronger baseline you had in mind.  We incorporated the state-of-the-art "Semi-Supervised Learning" approach by Wang et al (2020)—full citation above—in the initial rebuttal because it is the best we could find.  This paper is closest to our setting and yet it does not have any theoretical guarantees of asymptotic unbiasedness and valid confidence intervals, and we showed that it is biased and has invalid confidence intervals in our experiments. In general, bias and valid confidence intervals are theoretical properties that statisticians need to explicitly prove and no methods can be asymptotically unbiased and have valid confidence intervals by chance. We extensively reviewed the literature and know of no other method that has such guarantees. We are obviously now out of time to provide additional experiments in a rebuttal for you, but whether this is published at NeurIPS or elsewhere we are committed to getting this right, so if you have a citation or a method name in mind, please let us know and we are happy to incorporate an evaluation against it in the final version.
> > >
> > > **Response to Point (3)**
> > > To summarize the discussion thus far: the original review asked that we add F1 to the main results Table 2.  We promised to do so and in the rebuttal figure showed a snippet of Table 2 demonstrating that point.  We also emphasized that this is the only place F1 applies because the quantity of interest is again not the individual predictions but coefficients.  The reply was:
> > >
> > > *The F1 should also be presented in the method comparison results, given many of the data sets are multi-class classification and/or using imbalanced data sets.*
> > >
> > > We want to re-clarify that, for our main experiments, our target quantity is a coefficient (a vector of continuous variables), and thus, the final output is not a multi-class-classification. Therefore, regardless of whether the problem is imbalanced or multi-class, the final output is always a vector of continuous variables. Therefore, unfortunately, we do not have any F-1 score to report for the main results. But, as you suggested, we have already incorporated the F-1 score for the underlying LLM annotations (please see Table 2 in the PDF attached to the first rebuttal).

---

> > > ### Author Response · Authors · 2023-08-21
> > > **Response to Points (4-5)**
> > >
> > > **Response to Point (4)**
> > > To summarize the discussion thus far: The reviewer pointed out that the method is agnostic to the surrogate labels and suggested that the paper be rewritten to remove “so many specific mentions of LLMs and discussions about LLMs.” We offered to change the title and emphasize clearly in the paper the lack of dependence on LLMs.
> > >
> > > *The change in wording I believe is unsatisfactory and would like to continue arguing that the insistence on the importance of LLM is not representative and restricting the applicability of the method. I find this comment a bit misplaced: "you already build up a gold standard dataset with which you can use DSL to improve performance" the gold standard is needed anyway to start from and the other parts of the response mention that DSL's main goal was actually not to improve performance.*
> > >
> > > As we clarified above, we will change the title to "Using Imperfect Surrogates for Downstream Inference: Design-based Semi-supervised Learning for Social Science Applications of Large Language Models." Therefore, it is clear that LLMs are simply an application of the method. And, we do not restrict our paper to LLMs. Given that all of our experiments are about LLMs and in light of our point made in the paper and the rebuttal that many social scientists are moving towards LLMs, we think removing more reference to it could harm the ability of readers to find our paper.
> > >
> > > In response to the second point, given the extreme space constraint, we were brief here, but we mean (as argued repeatedly in the paper) that DSL improves over GSO (in terms of RMSE) and improves over SO and SSL (in terms of bias and coverage). When you say "the other parts of the response mention that DSL's main goal was actually not to improve performance." We think this is a misunderstanding again.  We were saying in the response that the goal is not to improve *performance of individual document classification* but instead to improve inference in the downstream regression. In the comment you are referencing we are talking about improving performance of the inference in the downstream regression.  We apologize for any confusion.
> > >
> > > **Response to Point (5)**
> > > To summarize the discussion thus far: The reviewer was concerned that we cited papers on accuracy claims that themselves don’t do enough to evaluate possible improvements in accuracy that could be achieved.  We thought this was part of the misunderstanding and clarified the specific goal we were after through the lens of one of the papers we cited.
> > >
> > > *The reference is primary about the 70% accuracy threshold copied from (Ziems at al 2023).*
> > >
> > > We agree that 70% accuracy claim in Ziems et al (2023) is not based on any particular evidence. But we show in the paper that, even when the accuracy of LLM classification (in the balanced case) is as high as 95%, the coverage is less than 80% (Figure 1-(a)). Therefore, our claims about the surrogate-only estimation is not dependent on Ziems et al (2023). As you suggested above, in the final version of the paper, we are happy to incorporate more experimental results where we vary the accuracy of LLM annotations.
> > >
> > > **Summary**
> > > We are concerned there may still be some confusion with you about the goal of our study and what semi-supervised learning entails in our setting.  We are more than happy to run benchmarks before publication (whether at NeurIPS or elsewhere) comparing to any baseline you like.

---

### Official Review · Reviewer_qD8b · 2023-07-10

**Soundness:** 3 good
**Presentation:** 3 good
**Contribution:** 2 fair
**Rating:** 6
**Confidence:** 4

**Summary:**

The authors consider using outputs of LLMs for labeling in downstream analysis in computational social science while guaranteeing statistical properties (asymptotic unbiasedness and uncertainty quantification). They show that directly using LLM-produced labels yields bias and invalid confidence intervals, then propose the design-based semi-supervised learning (DSL) estimator as an alternative. DSL combines surrogate labels from LLMs and gold-standard labels within a doubly robust procedure that guarantees valid inference even when the surrogates are arbitrarily biased and without requiring stringent assumptions. Experiments on 18 datasets show that DSL yields valid statistical inference while achieving error rates comparable to existing alternatives that focus on prediction with statistical guarantees.

**Strengths:**

The paper is very well written, the setting and methodology are well described and motivated. The assumptions and potential use cases are clearly stated. Further, though the proposed approach is conceptually simple, it blends sound ideas from semi-supervised learning (not necessarily in the, transductive, machine learning sense), doubly-robust estimation, K-fold cross-fitting and automated labeling with LLMs.

**Weaknesses:**

Perhaps the biggest weakness of the paper is that, though to the authors' admission, it prioritizes unbiasedness and coverage, however, DSL does not outperform GSO which is a very simple approach that does not require anything that is not available (a method for surrogate labels and the estimation of g). As a result, it is difficult to imagine a situation, more so in social science, where someone will prefer DSL over the far simpler GSO, unless the reviewer is missing something.

As minor points: i) the title of the paper is somewhat misleading because, as the authors point out, the proposed approach is agnostic to the methodology used to obtain the surrogate labels, so the fact they use LLMs is more of a detail of the experiments than a characteristic or contribution of the proposed approach; and ii) the paper (in its current form) does not seem a good fit for a conference proceeding considering the amount of important experimental details that need to be relegated to the supplementary material, without which, it is extremely difficult to properly understand the results.

**Questions:**

In several places, the authors emphasize the need for knowing \pi (the probability of obtaining a gold-standard label) and \pi > 0 for all samples. However, this is not further discussed or described in the experiments. For instance, in Section 2.2 in the Appendix, the authors claim that \pi is known because they can choose which samples to label, however how does that make \pi for all samples that are not going to be labeled (with probability 1) larger than 0?

**Limitations:**

The authors discuss some of the limitations of the proposed approach. However, the limitations are quite generally about the addressed setting and less so about the methodology. Also, they point to four limitations, however, only three are described (the particular setting, access to gold standard labels and the focus on bias and coverage rather than MSE).

---

> ### Author Rebuttal · Authors · 2023-08-08
>
> Thank you for the kind words about the paper! We are glad that you found the setting and methodology well-described and motivated.
>
>
> **(1)** *Perhaps the biggest weakness of the paper is that [...] it prioritizes unbiasedness and coverage, however, DSL does not outperform GSO which is a very simple approach that does not require anything that is not available[...]. As a result, it is difficult to imagine a situation [...] where someone will prefer DSL over the far simpler GSO, unless the reviewer is missing something.*
>
> As the reviewer notes, gold standard labels only (GSO) and DSL are both asymptotically unbiased and have valid confidence intervals. In terms of the RMSE, we now see that Figure 2 did not convey precise information about how much gains DSL can provide compared to GSO. We clarify here that DSL indeed provides substantial RMSE gains over GSO. As written in the original draft, in our Congressional Bills Data experiment, we find that the RMSE gain by DSL over GSO is about 10% with zero-shot learning LLMs and about 30% with five-shot learning. To make these numbers more intuitive, we translated the RMSE improvement in terms of the number of hand-coded documents, i.e., how many more hand-coded documents does GSO need to achieve the same RMSE as DSL? This translation is possible because both estimators are asymptotically unbiased, and all the gains come from a smaller variance. We find that, for the zero-shot case, the RMSE gain by DSL is equivalent to having 50% more hand-coded documents, and for the five-shot case, the RMSE gain by DSL is equivalent to having 100% more hand-coded documents. Given the high cost of obtaining hand-coded documents, this shows that computational social scientists can obtain unbiased estimates and valid confidence intervals with much fewer hand-coded documents when they use DSL instead of GSO. We provide a new Figure R-1 (see PDF) based on our experiment.
>
> We also emphasize that as the quality of LLMs improves and their deployment gets easier, the cost-benefit proposition for DSL over GSO will be even greater. In Figure R-2 of the attached PDF, we show the degree of gains of DSL for higher accuracies in simulated data.
>
> **(2)** *As minor points: i) the title of the paper is somewhat misleading because, as the authors point out, the proposed approach is agnostic to the methodology used to obtain the surrogate labels, so the fact they use LLMs is more of a detail of the experiments than a characteristic or contribution of the proposed approach;*
>
> Following your and other reviewers’ advice, we are changing the title to **“Using Imperfect Surrogates for Downstream Inference: Design-based Semi-supervised Learning for Social Science Applications of Large Language Models.”** We will also emphasize the generality of DSL and clarify that LLMs are one application in the revision. Please see more detailed responses in our Global Response (at the top of this page).
>
> **(3)** *[...] the paper (in its current form) does not seem a good fit for a conference proceeding considering the amount of important experimental details that need to be relegated to the supplementary material, without which, it is extremely difficult to properly understand the results.*
>
> We will move more experimental details out of the supplement (in particular, we will be more clear about the specification of downstream analyses, the details of LLM annotations, and training procedures). We are open to suggestions about which would be most useful to move. While we know that many experimental details are in the appendix, we see the experiments as primarily the validation that the theory holds in practice and hence not the main focus. We think NeurIPS would be an excellent venue because this paper tries to address the intersection and frontier of computational social sciences: as some reviewers note, the common social science task of making downstream statistical analysis (and their focus on bias and coverages) is relatively new to computer science communities, while the valid use of LLMs is new to social scientists. Given this novelty and the interdisciplinary nature of our work, we believe NeurIPS is the best venue.
>
> **(4)** *[...] the authors emphasize the need for knowing \pi (the probability of obtaining a gold-standard label) and \pi > 0 for all samples. However, this is not further discussed or described in the experiments. [...] [T]he authors claim that \pi is known because they can choose which samples to label, however how does that make \pi for all samples that are not going to be labeled (with probability 1) larger than 0?*
>
> Thank you for your question. $\pi$ is the probability of being sampled for hand-coding. As an example, consider random sampling. If I have 10000 documents and I sample 100 of them to hand annotate completely at random, $\pi$ = 1/100 for all documents regardless of whether any individual document is chosen or not. Therefore, in the social science applications where researchers can sample documents for hand-coding, this $\pi$ is known, and it is greater than 0. In our experiment, to show the generality of our approach, we use the stratified sampling procedure where we stratified based on LLM labels such that we can oversample rare cases when we conduct hand-coding. We thank the reviewer for this clarification question, and we will add a simple example in the paper itself to hit this point home and will reinforce it when we describe applications.
>
> **(5)** *The authors discuss some of the limitations of the proposed approach. However, the limitations are quite generally about the addressed setting and less so about the methodology. Also, they point to four limitations, however, only three are described [...].*
>
> Thank you also for flagging the concerns about the limitations, we will expand this (and get the count correct!) to focus more on limitations specific to the method with an emphasis on points raised by the reviewers.

---

> > ### Comment · Reviewer_qD8b · 2023-08-16
> >
> > Thanks for the very detailed response (including the figures) to the weaknesses and questions raised in the original review. The answers to points (1) and (4) will be specially important to address in the revision to make it easier for the reader. I have updated my score accordingly.

---

> > > ### Author Response · Authors · 2023-08-16
> > >
> > > Thank you so much for your kind words. In the revised manuscript, we will make sure to incorporate these changes, especially (1) and (4), as you suggested. Thank you again for your careful engagement with our work!

---

### Author Rebuttal · Authors · 2023-08-08

Thank you to all the reviewers for the careful engagement with our work. In this paper, we proposed how to use imperfect LLM annotations in downstream regression analyses, while guaranteeing asymptotic unbiasedness and proper uncertainty quantification, which are fundamental to social science research.

All four reviewers agreed that the problem and setting of our paper are interesting, and our proposed methodology has strong theoretical guarantees. Most felt that our experiments using 18 different computational social science data sets were strong as well. In this reply, we address some of the common critiques and provide a guide to the new figures in the attached PDF.

**Why Bias and Coverage are Important in the Social Sciences**

We want to further clarify why we focused on bias and valid uncertainty quantification and why they are often prioritized over RMSE in the social sciences. This is primarily because social scientists often conduct statistical analyses to *test social science hypotheses* rather than making predictions. For example, when studying online hate speech, social scientists would be interested in testing whether highly educated people are less likely to post hate speech. To conduct such hypothesis testing, valid uncertainty quantification (with confidence intervals and corresponding p-values) is essential. If there exist several estimators that are asymptotically unbiased and provide valid confidence intervals, it is best to choose the one with a lower RMSE. In this paper, we showed that the gold-standard only (GSO) estimator and DSL estimator are the only two methods that are unbiased and have valid confidence intervals, and DSL outperforms GSO in terms of RMSE, which we elaborate more on below.

**Concerns about RMSE Improvements over Existing Alternatives**

There was concern about the RMSE improvements of DSL over the gold standard only (GSO) and semi-supervised learning (SSL) benchmarks.

In terms of comparisons against GSO, we admit that Figure 2 did not convey clear information about how much DSL can improve RMSE (this was masked because of a log scale). Please let us re-clarify this point here. As written in the original draft, in our Congressional Bills Data experiment, we find that the RMSE gain is about 10% with zero-shot learning LLMs and about 30% with five-shot learning. To make these numbers more intuitive, we translated the RMSE gains into the gains in the number of hand-coded documents, i.e., how many more hand-coded documents does GSO need to have the same RMSE as DSL? This translation is possible because both estimators are asymptotically unbiased, and all the gains come from a smaller variance. We find that, for the zero-shot case, the RMSE gain by DSL is equivalent to having 50% more hand-coded documents, and for the five-shot case, the RMSE gain by DSL is equivalent to having 100% more hand-coded documents. Given the significant cost of obtaining hand-coded documents, this shows that computational social scientists can obtain unbiased estimates and valid confidence intervals with much fewer hand-coded documents when they use DSL instead of GSO. We provide a new Figure R-1 (see PDF) based on our experiment. We also provide a new simulation result in Figure R-2 to show that this sample size gain increases as the accuracy of LLMs increases. We will incorporate these new results and clarification in the next revised version.

In terms of comparison against SSL, we want to first re-emphasize that SSL does not meet the social science priority of bias and coverage. Following the reviewers’ comments, we added several additional variants of SSL methods, and all of them are still biased and have invalid confidence intervals, as our theory suggested. Focusing only on RMSE, we expect that SSL, which is optimized for RMSE, can outperform DSL. However, in our experiment, we showed that DSL, while maintaining unbiasedness and valid confidence intervals, can achieve RMSE comparable to SSL (even though it is often higher than SSL in a finite sample). Importantly, we also show that, as sample size increases, DSL provably dominates SSL in terms of RMSE because bias of SSL does not vanish with sample size, while variance of DSL will vanish with o_p(1/n). One reviewer raised concerns over the strength of the SSL baseline. We have added our SSL baselines, including an estimator by Wang et al. (2020), which, as far as we know, is the only published article that proposes an estimator that covers exactly the same settings as ours (downstream regression analyses). As it requires stronger parametric assumptions, it is, in general, biased and has invalid confidence intervals in our experiments (see Figure R-3 in the PDF).

**Concerns about the Title**

Multiple reviewers raised concerns about the title because DSL can be applied to any surrogates and is not limited to LLMs. To address this, we are going to change the title to: “**Using Imperfect Surrogates for Downstream Inference: Design-based Semi-supervised Learning for Social Science Applications of Large Language Models.**” We will also emphasize the generality of DSL and downplay the specific LLM angle in the paper. But we hope to keep some of the framings for three reasons. First, LLMs are becoming a huge part of the computational social sciences (e.g., Ziems et al., 2023), and the application to LLMs will be more than 90% of the use cases of our approach. Second, we think the approach works very well with the applied zero/few-shot LLM workflow. You have some LLM annotations, but you have to sample some observations to check that it is working; in doing so, you already build up a gold standard dataset with which you can use DSL to improve performance. Third, we think LLMs will continue to get better, and that will just make the DSL estimator improve even more.

**Additional Changes**

Additional changes are specific to individual reviewers and discussed in those rebuttals.

---

### Decision · Program_Chairs · 2023-09-21

**Decision:**

Accept (poster)

**Comment:**

The paper proposes a strategy for unbiased estimation of regression coefficients using imperfect surrogate labels.

In the initial reviews there was some confusion, much of which the authors have adequately clarified with good answers. One concern raised, which I agree with, centers on the heavy emphasis the authors put on using large language models as a source of surrogate labels, even if the method works with many other sources of surrogate labels. The authors have committed to reducing this emphasis and updating the title, which I strongly urge them to follow through with.

Overall the method and results presented here are of interest to the community, and I recommend acceptance.